# The Relationship Between the Phonological Processing Network and the Tip-of-the-Tongue Phenomenon: Evidence from Large-Scale DTI Data

**DOI:** 10.3390/bs15070977

**Published:** 2025-07-18

**Authors:** Xiaoyan Gong, Ziyi He, Jun Wang, Cheng Wang

**Affiliations:** 1School of Psychology, Zhejiang Normal University, Jinhua 321004, China; gongxy@zjnu.edu.cn (X.G.); heziyi0420@163.com (Z.H.); jun.wang@zjnu.edu.cn (J.W.); 2Key Laboratory of Intelligent Education Technology and Application of Zhejiang Province, Zhejiang Normal University, Jinhua 321004, China

**Keywords:** tip-of-the-tongue phenomenon, phonological processing, DTI, brain connectivity, graph theory

## Abstract

The tip-of-the-tongue (TOT) phenomenon is characterized by a temporary inability to retrieve a word despite a strong sense of familiarity. While extensive research has linked phonological processing to TOT, the exact nature of this relationship remains debated. The “blocking hypothesis” suggests that the retrieval of target words is interfered with by phonological neighbors, whereas the “transmission deficit hypothesis” posits that TOT arises from insufficient phonological activation of the target words. This study revisited this issue by examining the relationship between the microstructural integrity of the phonological processing brain network and TOT, utilizing graph-theoretical analyses of neuroimaging data from the Cambridge Centre for Ageing and Neuroscience (Cam-CAN), which included diffusion tensor imaging (DTI) data from 576 participants aged 18–87. The results revealed that global efficiency and mean degree centrality of the phonological processing network positively predicted TOT rates. At the nodal level, the nodal efficiency of the bilateral posterior superior temporal gyrus and the clustering coefficient of the left premotor cortex positively predicted TOT rates, while the degree centrality of the left dorsal superior temporal gyrus (dSTG) and the clustering coefficient of the left posterior supramarginal gyrus (pSMG) negatively predicted TOT rates. Overall, these findings suggest that individuals with a more enriched network of phonological representations tend to experience more TOTs, supporting the blocking hypothesis. Additionally, this study highlights the roles of the left dSTG and pSMG in facilitating word retrieval, potentially reducing the occurrence of TOTs.

## 1. Introduction

The tip-of-the-tongue (TOT) phenomenon is a frustrating state in which speakers feel highly familiar with a word but are temporarily unable to retrieve it ([16]). TOT states occur in people’s daily lives. This phenomenon happens more frequently in elderly adults and individuals with conditions such as dyslexia, aphasia, or Parkinson’s disease ([17]; [29]; [76]). Uncovering the mechanisms underlying TOT states can help us reduce their occurrence and improve life quality. Extensive research indicates that retrieving words from the mental lexicon involves two sequential processes, beginning with the identification of a lexical concept that accurately conveys the intended meaning and followed by the retrieval of the phonological word form that represents this concept ([63]; [57]; [56]). The inability to complete the second step is often cited as the primary cause of TOT states ([20]; [78]; [67]). Among the factors contributing to phonological retrieval failure, the speaker’s lexical phonological network plays a significant role ([78]). However, two distinct patterns of findings regarding the relationship between this network and TOT occurrences have created a longstanding debate in the field. This controversy has given rise to two contrasting hypotheses: the blocking hypothesis and the transmission deficit hypothesis.

The blocking hypothesis of TOT proposes that dense phonological neighbors can obstruct the retrieval of intended targets, resulting in TOTs. Early researchers ([93]; [94]) observed that in everyday life, when stuck in a TOT state, individuals often produced “interlopers” (or “blocking words”) that sound similar to the target word. Therefore, researchers hypothesized that the interlopers might interfere with the retrieval of the target words, resulting in TOTs. A systematic diary study ([73]) supports this hypothesis by revealing that in 70% of the naturally occurring TOTs, participants reported blocking words with high phonological similarity to the target words. Further evidence was provided by [49] ([49]) in which a definition naming task in combination with a priming paradigm was used to induce TOTs. Directly after reading the definition of a target word, participants were presented with an interloping word, which was phonologically related, semantically related, or unrelated to the target word. The results show that phonologically related interlopers were more likely to induce TOT states, compared to unrelated or only semantically related interlopers. The researchers explained that the priming effect facilitated the erroneous retrieval of interlopers, which disrupted the phonological retrieval of the target words ([49]). These findings aligned with the spreading activation model of speech production ([25]), which suggests that closely linked similar items can influence each other through a spreading activation effect. When a target word is to be activated, activation spreads to similar competitors. Any factors that increase the activation levels of these competitors can lead to retrieval failures, as is the case with TOTs. This hypothesis is further supported by recent bilingual studies, which reported that bilinguals experienced more TOTs than monolinguals, probably due to extra phonological interference from other language systems ([84]). Additionally, word retrieval has been shown to be slowed by phonological neighbors in a blocked cyclic naming task ([14]; [71]). In this task, participants were asked to name pictures within a “homogeneous” block, where the picture names shared phonological overlap (e.g., cat, cap, or cab), or a “heterogeneous” block, in which the picture names had no phonological overlap. The key finding was that response latencies were slower in the homogeneous condition compared to the heterogeneous condition, indicating the presence of interference from phonological neighbors ([71]). Furthermore, TOT rates are higher in older than in younger adults ([21]; [78]). The blocking hypothesis explains this age effect by suggesting that aging weakens the inhibitory ability of old adults, making them more susceptible to interferences ([98]), or that vocabulary or lexical knowledge increases with age, resulting in stronger competition from neighbors ([55]; [86]).

The transmission deficit hypothesis (TDH) offers an alternative explanation for TOTs, arguing that retrieval failure occurs not because of interference from interlopers, but due to insufficient phonological activation of target words caused by weakened connections between the semantic nodes and the phonological nodes or connections within the phonological nodes ([60]; [20]; [45]). Proponents of TDH ([63]; [45]) criticize [49] ([49]) for claiming that their priming effect of interlopers could be explained by a strategic retrieval effect, rather than interference from the interloper, because their primes follow the presentation of the definition, by which time lexical access may have well been completed. To avoid this, [45] ([45]) adopted a repetition priming paradigm to eliminate the strategic retrieval effects, they observed a result pattern very opposite to Jones and Langford’s, i.e., phonologically related prime words significantly reduced TOT occurrences, indicating that processing phonologically related prime words activated shared phonological representations, compensated for the transmission deficit, which in turn facilitated the successful retrieval of the target word (see [1] for a similar result). Moreover, several other studies also support the TDH by demonstrating that phonological neighbors reduce TOT occurrences. [33] ([33]) found that bilinguals experienced fewer TOTs with cognates—words that share similar meanings and phonological forms—compared to non-cognates. [89] ([89]) showed that higher phonological neighborhood density was associated with lower TOT rates among younger adults, because more phonological neighbors strengthen connections between phonological nodes. Additionally, research has also shown that prior production of homophones can significantly reduce TOT rates for older adults ([19]). Together, these findings highlight the role of enhanced phonological activation in mitigating TOTs, further supporting the TDH.

The controversy between the blocking hypothesis and the TDH remains unresolved, largely due to variations in choices of methodology and inconsistent results across different studies ([49]; [45]). Additionally, the limited sample sizes in these studies—often comprising a small number of college students or older volunteers—may not adequately represent the general population. The current study aims to address the ongoing debate by examining individuals’ lexical phonological network as a key factor. The blocking hypothesis suggests that target words within a more enriched phonological network face increased competition and interference from denser phonological neighbors, leading to higher TOT occurrences. In contrast, TDH posits that individuals with more enriched phonological networks possess a greater number of connections among their phonological neighbors, resulting in enhanced activation of target words and consequently reduced TOT occurrences. The current study utilizes the microstructural integrity of the brain’s phonological processing network as a proxy for individuals’ richness of the lexical phonological network and investigates how this microstructural integrity affects TOT occurrences. Using this proxy is justified, as previous research has established that higher anatomical integrity within the language network is associated with larger vocabulary size ([59]; [41]; [4]; [26]), suggesting that a higher integrity of the phonological processing brain network represents a more enriched lexical network of phonological representations ([32]; [75]).

According to meta-analyses of speech production ([39]; [43]; [44]; [51]; [66]), lexical phonological processing involves a network of multiple brain regions, which consists of the bilateral superior temporal gyrus (STG) including dorsal and posterior part of it (dSTG and pSTG), the left posterior supramarginal gyrus (pSMG), the left anterior insula (aINS), the left posterior inferior frontal gyrus (pIFG), and the left premotor (also called Precentral in HOA-112, PMA). The bilateral STG is responsible for storing phonological representations, while the left pSMG plays a role in the temporary storage of phonological representations ([69]). Furthermore, the left pIFG and premotor area are frequently implicated in tasks related to processing phonological information, such as syllabification and sequencing ([27]; [34]; [39]). Additionally, the left aINS is believed to be involved in phonological retrieval ([78]). This is predominantly left-lateralized.

Distinct regions within the brain network communicate via white matter tracts that interconnect them ([58]). White matter tracts serve as the anatomical foundation for functional connectivity between gray matter regions ([11]), and damage to white matter can significantly impair cognitive functions. For example, injury to the superior longitudinal fasciculus has been shown to play a crucial role in severe language dysfunction ([28]). Conditions such as neurological disorders and aging can significantly alter the structural connectivity of the brain network ([12]; [22]; [65]; [74]; [76]; [80]; [83]). DTI is a technique that images the diffusion of water molecules in brain tissue, enabling the mapping of white matter tracts and the assessment of neural structural connectivity ([8]). Previous studies have revealed that regions critical for phonological processing, such as the IFG, STG, SMG, PMA, and INS, communicate through white matter tracts like the arcuate fasciculus ([23]), superior longitudinal fasciculus ([50]), and extreme capsule ([62]), forming a complex neural network critical for speech production. Graph theoretical analysis provides a robust framework for characterizing the topological characteristics of complex neural networks. This approach quantifies efficiency, modularity, and centrality through global and nodal metrics ([9]; [18]; [75]; [82]), thereby providing a quantitative measure for estimating the microstructural integrity of neural networks and identifying key nodes and pathways that play crucial roles within these networks. Employing this approach has enabled researchers to gain insights into how the integrity of neural networks relates to cognitive processing and behavioral outcomes in various neurological contexts ([10]; [30]). For example, studies have indicated that the diminished naming performance of Parkinson’s patients is associated with decreased network metrics such as global efficiency, local efficiency, clustering coefficient, and degree centrality ([31]; [76]).

The current study aims to assess the relationship between lexical phonological processing abilities and TOT occurrences, using the microstructure integrity of the phonological brain network as a proxy for individuals’ lexical phonological processing abilities. Microstructural integrity was assessed through a graph theory approach applied to DTI data from a large-scale cross-sectional cohort (Cam-CAN, ages 18–87, N = 576) ([79]; [87]). This dataset also includes a behavioral measure of TOT rates, which were collected through a public figure naming task. This task is a well-established method for eliciting TOT states and has been commonly used in various studies ([96]; [20]; [35]; [68]). [20] ([20]) showed that attempts to retrieve a celebrity’s name produced TOT experiences in about 60% of trials, highlighting the task’s propensity to induce TOT states. As individuals’ phonological processing abilities decline due to disease or brain impairment, they may experience increased difficulty in retrieving phonological representations ([31]; [76]). If the richness of a lexical phonological network, as indicated by the integrity of the phonological processing network, demonstrates a positive correlation with the frequency of TOT states, the evidence would align with the blocking hypothesis. Conversely, if the findings reveal a negative correlation between these variables, this would lend support to the TDH.

## 2. Methods

Data used in the preparation of this study were obtained from the Cam-CAN data repository (available at https://cam-can.mrc-cbu.cam.ac.uk/dataset/) ([79]; [87]). The protocol for the Cam-CAN study has been approved by the institution’s ethics committee. Specifically, we analyzed a subset of the dataset that includes the TOT task, DTI data, and demographic information on the participants. We obtained permission to use the data on 30 August 2023.

### 2.1. Participants

After confirming that all participants completed the required tasks for this study, their MRI images were visually inspected for quality. Six datasets were excluded due to poor image quality or missing data files, resulting in a final sample of 576 participants, including 292 males and 284 females, aged 18 to 87 years (*M* = 55.09, *SD* = 17.77). All participants were assessed as cognitively healthy, with a Mini-Mental State Examination (MMSE) score of 24 or higher. There was no evidence of organic pathology, psychiatric disorders, or a history of substance abuse among the participants ([79]; [87]).

### 2.2. Behavioral Data

TOT rates were collected using a public figure naming task ([79]), which included 50 faces of famous individuals from various fields (e.g., actors, musicians, and politicians). The pictures of these faces were presented in a pseudorandom order. Each trial began with a 1000 ms fixation, which was then replaced by a picture of a face that remained on the screen for 5000 ms. Participants were instructed to name each person upon seeing the picture. The task allowed for three types of responses: “Know” (participants correctly named the person), “Don’t Know” (participants did not know the name of the person), and “TOT” (participants were confident that they knew the name but could not retrieve it). The TOT rate was calculated using the following formula: TOT rate=TOT/(TOT+Know), where TOT represents the number of “TOT” responses and Know represents the number of “Know” responses.

### 2.3. Data Acquisition

The imaging data retrieved for this study comprised diffusion- tensor imaging (DTI) data and high-resolution T1-weighted structural images. DTI data was acquired using a twice-refocused spin–echo sequence, with 30 diffusion gradient directions for each of two b-values—1000 and 2000 s/mm^2^—plus three images acquired with a b-value of 0. Other parameters included Repetition Time (TR) = 9100 ms, Echo Time (TE) = 104 ms, FOV =192×192 mm, voxel size =2×2×2 mm, 66 axial slices, number of averages = 1, and acquisition times of 10 min and 2 s. A high-resolution 3D T1-weighted structural image was acquired using a Magnetization Prepared Rapid Gradient Echo (MPRAGE) sequence with the following parameters: FOV = 256×240×192 mm, voxel size = 1×1×1 mm, GRAPPA acceleration factor = 2; total acquisition time = 4 min and 32 s; TR/TE/TI = 2250 ms/2.99 ms/900 ms, respectively; and flip angle = 9°.

### 2.4. Image Preprocess

All preprocessing steps were conducted using the FMRIB Software Library (FSL 6.0.6.5) (https://fsl.fmrib.ox.ac.uk/fsl/fslwiki/FSL, accessed on 13 April 2025) ([48]). The T1-weighted structural images were preprocessed with the following steps: (1) reorientation and cropping to remove the cervical region, (2) skull stripping using the Brain Extraction Tool (BET) ([81]), (3) registration to the MNI152 space using both affine transformation (FLIRT) ([47]) and non-linear transformation (FNIRT) ([5]), resulting in the generation of warp field files. The DWI data were preprocessed with the following steps: (1) correction for head motion and eddy currents ([6]), (2) removal of non-brain voxels using the BET ([81]), (3) co-registration to the T1 structural space using rigid transformation (FLIRT) ([47]), and (4) registration to the MNI152 space by applying the warp field files ([5]).

### 2.5. Network Construction

**ROI selection.** Based on previous meta-analyses ([39]; [43]; [44]; [51]; [66]), eight regions of interest (ROIs) responsible for lexical phonological processing in speech production were selected: the bilateral superior temporal gyrus (STG), specifically the dorsal and posterior parts (dSTG and pSTG), the left posterior supramarginal gyrus (pSMG), the left anterior insula (aINS), the left posterior inferior frontal gyrus (pIFG), and the left premotor area (PMA). The Harvard Oxford atlas (HOA-112) was used to parcellate the cerebral cortex into 112 predefined cortical regions ([61]), and the masks corresponding to the eight ROIs were selected. Because subregions of the INS are not defined in HOA-112, we used the ROI drawing feature of FSL to accurately outline its anterior part, based on the anterior–posterior partition boundaries of the INS ([7]). Table 1 lists the coordinates of all ROIs. All ROI masks were generated using the FSL and the NumPy ([36]) and nibabel ([15]) libraries in Python 3.9.6.

**Probabilistic tracking.** To construct the structural network, DTI data were further processed using bedpostX ([38]) to build up distributions of diffusion parameters at each voxel. Subsequently, probabilistic fiber-tracking runs by using probtrackx2 ([37]) to generate a histogram of the number of streamlines connecting specific brain regions. The “multiple masks” option of probtrackx2 was applied, with all other settings maintained at their default values. Consequently, a connectivity matrix was constructed, with each element representing the number of streamlines from a seed ROI to a target ROI.

### 2.6. Graph Analysis

The brain is abstracted into mathematical concepts in the form of networks, reflecting its real-world significance. After probabilistic tracking, we constructed the structural connectivity (SC) matrix for the phonological processing network, which involved defining brain regions as nodes and the white matter fiber tracts connecting these regions as edges. In this study, ROIs were defined as nodes *i* and *j*, with the average number of fibers between nodes *i* to *j* and *j* to *i* considered as edges. The weight of each node pair $w_{ij}$ was normalized using the corresponding elements of each matrix, considering the varying network costs among participants ([90])$$w_{ij}=\frac{\log\left(p_{ij}\right)-\min{\left\{\log\left(P_{ij}\right)\right\}}_{1\leq i\neq j\leq N}}{\max{\left\{\log\left(P_{ij}\right)\right\}}_{1\leq i\neq j\leq N}-\min{\left\{\log\left(P_{ij}\right)\right\}}_{1\leq i\neq j\leq N}}$$
where *N* is the total number of nodes (1≤i≠j≤N).

In this study, we utilized GRETNA 2.0.0 (https://www.nitrc.org/org/projects/gretna, accessed on 13 April 2025) ([91]) and the Brain Connectivity Toolbox ([75]) to compute all graph theoretical metrics. Our focus was on measuring the microstructural integrity of the network through both global characteristics (local efficiency, global efficiency, mean degree centrality, and mean clustering coefficient) and local characteristics (degree centrality, nodal efficiency, and clustering coefficient) of structural connectivity ([75]).

Global efficiency measures the efficiency of parallel information transmission in a network, calculated as the average of the reciprocals of the characteristic path lengths of all nodes ([75]). The formula is$$E_{global}=\frac{1}{n}\sum_{i\in N}\frac{\sum_{j\in N,j\neq i}{\left(d_{ij}^{w}\right)}^{-1}}{n-1}$$
where *N* represents the set of all nodes in the network and *n* is the number of nodes, $d_{ij}^{w}$ is the shortest weighted path length between node *i* and *j*. The range of values for global efficiency ranges from 0 to 1, where a larger value indicates higher global efficiency. Global efficiency reflects the efficiency of long-distance information transmission in the entire network ([18]).

The local efficiency of a network is a measure of the efficiency of information exchange within each subgraph ([2]). It is obtained by averaging the local efficiencies of all nodes within the network. The formula is$$E_{local}=\frac{1}{2}\sum_{i\in N}\frac{\sum_{j,h\in N,j\neq i}{\left({w_{ij}w_{ih}[d_{jh}^{w}(N_{i})]}^{-1}\right)}^{1/3}}{k_{i}(k_{i}-1)}$$
where the meanings of *N* and n align with global efficiency, while $d_{jh}^{w}\left(N_{i}\right)$ represents the length of the weighted shortest path between nodes *j* and h that includes only the neighbors of node *i*. The $k_{i}$ denotes the number of links connected to node *i*. Local efficiency shows how well information can still flow between the remaining nodes in a network when one node is removed. This helps us understand how resilient the network is to failures ([54]).

Nodal efficiency is a measure of network integration, reflecting a node’s ability to transmit information to all other nodes in the network. In weighted graphs, nodal efficiency is calculated as the reciprocal of the shortest path length between one node and another, accounting for the weights of all edges that connect them ([53]). The formula is$$E_{i}=\frac{\sum_{j\in N,j\neq i}{\left(d_{ij}^{w}\right)}^{-1}}{n-1}$$
where *N* represents sets of all nodes in the network and *n* is the number of nodes. The node with high nodal efficiency could communicate and share information with other nodes easily.

Degree centrality quantifies the number of connections a node has with other nodes in the network, reflecting its importance within the network ([18]). In a weighted network, degree centrality is defined as the sum of all weights associated with node *i* and its neighboring nodes ([75]). The formula is$$k_{i}^{w}=\sum_{j\in N}w_{ij}$$
where the $w_{ij}$ represents the weight between node *i* and *j*. The degree centrality has a straightforward neurobiological interpretation: nodes with high degrees interact with many other nodes in the network both structurally and functionally ([75]). At the global level, the average strength, also known as the average degree centrality, denotes the average value of the strength (degree centrality) of all nodes within a network ([75]). This metric captures the connectivity of the entire weighted network. A higher average strength indicates greater connectivity among the nodes.

The clustering coefficient describes the local processing efficiency of a network. When two neighbors of a node are connected to each other, they form a closed triangle with that node. The clustering coefficient is calculated as the ratio of the number of triangles around a node relative to the number of possible triangles that could be formed around that node ([92]). The formula is$$C^{w}=\frac{1}{n}\sum_{i\in N}\frac{{2t}_{i}^{w}}{k_{i}(k_{i}-1)}$$
where $t_{i}^{w}$ represents the geometric mean of triangles surrounding node *i*. A high clustering coefficient typically indicates that a node has closer connections with its neighbors, leading to increased local efficiency and synergistic effects ([18]). At the global level, the mean clustering coefficient refers to the mean clustering coefficient across all nodes within a network ([75]). This metric quantifies the tendency of nodes within the network to form tightly interconnected clusters, providing insights into the local efficiency and modular organization of the network. A higher average clustering coefficient suggests a greater propensity for nodes to form local clusters, indicating enhanced segregation and localized interactions within the network.

### 2.7. Statistical Analysis

Statistical analysis was conducted using R (version 4.3.2). To examine the relationship between graph-theoretical metrics of the phonological processing network and TOT rates, we performed separate multiple linear regression analyses for each network metric. Each model included one metric as the predictor, with TOT rates as the dependent variable. Gender and age were accounted for as covariates. For nodal metrics, each model focused on a specific metric, utilizing the metric calculated from all ROIs as the predictor. The bidirectional stepwise regression method was employed to select important predictors based on the Akaike Information Criterion (AIC) ([3]). Multicollinearity was controlled by ensuring that the VIF remained below 3.

## 3. Results

### 3.1. Global Properties

To investigate the relationship between the global properties of the phonological network and the TOT rate, we conducted a series of multiple regression analyses, separately for each global metric. The results (see Table 2) show that global efficiency (*t* = 2.320, *p* = 0.021) and mean degree centrality (*t* = 1.925, *p* = 0.055) positively predict TOT rates in their respective models. In contrast, local efficiency (*t* = −0.516, *p* = 0.606) and mean clustering coefficient (*t* = 1.925, *p* = 0.654) were not significant predictors of TOT rates. In all models, age emerged as a significant predictor of TOT rates (*t*s > 8.331, *p*s < 0.001), while gender did not show significant effects (*t*s < −0.034, *p*s > 0.001). As the global metrics represent the richness of the lexical phonological network, these positive correlations indicate that a more enriched phonological lexical network can impede word retrieval and lead to an increased occurrence of TOTs.

### 3.2. Nodal Properties

To investigate the node-specific contributions to the tip-of-the-tongue (TOT) rate, we conducted stepwise regression analyses separately for each nodal metric, with age, gender, and all ROIs as predictors. See Table 3 and Figure 1 for results. For nodal efficiency, both the left pSTG (*t* = 2.708, *p* = 0.007) and the right pSTG (*t* = 2.655, *p* = 0.008) significantly positively predicted TOT rates. For degree centrality, the left dSTG (*t* = −2.191, *p* = 0.029) significantly negatively predicted TOT rates. Lastly, for clustering coefficient, the left PMA (*t* = 3.091, *p* = 0.002) significantly positively predicted TOT rates, whereas the left pSMG (*t* = −2.814, *p* = 0.005) significantly negatively predicted TOT rates. The positive correlations of the bilateral pSTG and left PMA with TOT rates indicate that phonological neighbors interfere with target word selection, whereas the negative correlations of the left pSMG and left dSTG with TOT rates suggest that competition resolution and self-monitoring facilitate word selection.

### 3.3. Age-Related Differences

As age significantly contributed to TOT rates in all regression models, we further examined how age affects the relationship between the network properties of the phonological processing network and TOT rates. To do this, we first divided participants into three groups: young adults (18–34 years), middle-aged adults (35–65 years), and elderly adults (over 65 years) ([52]). We then conducted multiple regression analyses separately for each age group. The results of global metrics indicated that only the elderly adult group exhibited a significant correlation between TOT rates and network metrics (see Table 4). Specifically, global efficiency (*t* = 2.756, *p* = 0.006), local efficiency (*t* = 2.126, *p* = 0.035), mean degree centrality (*t* = 2.555, *p* = 0.011), and mean clustering coefficient (*t* = 2.555, *p* = 0.011) positively predicted TOT rates.

For each nodal metric, we conducted stepwise regression analyses in each age group, with gender and all ROIs as predictors (see Table 5 for results). For nodal efficiency, in young adults the left pSTG positively predicted TOT rates (*t* = 2.779, *p* = 0.007), whereas the left dSTG negatively predicted TOT rates (*t* = −2.170, *p* = 0.033); in middle-aged adults, no predictor reached significance (*t*s < 1.68, *p*s > 0.09); in elderly adults, the left PMA positively predicted TOT rates (*t* = 2.860, *p* = 0.005). For degree centrality, in young adults, the left pSTG positively predicted TOT rates (*t* = 2.077, *p* = 0.041), whereas the left dSTG and gender were non-significant (*t*s < 1.86, *p*s > 0.06); in middle-aged adults, no significant effects emerged (*t*s < 1.64, *p*s > 0.10); in elderly adults, the left PMA positively predicted TOT rates (*t* = 2.796, *p* = 0.006). For clustering coefficient, in young adults, no predictor was significant (*t*s < 1.45, *p*s > 0.15); in middle-aged adults, the left dSTG negatively (*t* = −1.976, *p* = 0.049) and the left PMA positively (*t* = 2.523, *p* = 0.012) predicted TOT rates; in elderly adults, the left PMA (*t* = 2.053, *p* = 0.041) positively predicted TOT rates while the left pSMG negatively predicted TOT rates (*t* = −2.721, *p* = 0.007).

In summary, age influenced the relationship between the network metrics of the phonological network and TOT rates, with effects observed in the full sample analysis being restricted to one or two age groups. However, the directions of these effects were highly consistent between the full sample analysis and the age-based group analysis.

## 4. Discussion

The current study investigated the relationship between the phonological network and TOT states, utilizing the microstructural integrity of the brain’s phonological processing network as a proxy for the lexical phonological network to examine the occurrences of TOT. Our findings revealed that global efficiency and mean degree centrality positively predicted TOT rates, suggesting that a more enriched lexical phonological network may exacerbate competition by increasing the accessibility of phonological neighbors, as proposed by the blocking hypothesis. At the nodal level, the bilateral pSTG demonstrated significant positive predictions of TOT for nodal efficiency, while the left dSTG exhibited negative predictions in both nodal efficiency and degree centrality. Additionally, the left PMA showed significant positive associations with TOT in both degree centrality and clustering coefficient, whereas the left pSMG revealed a significant negative prediction in clustering coefficient. Collectively, these findings challenge the core premise of the TDH—that TOT arises due to weakened transmission pathways—and instead lend support to the blocking hypothesis, which posits that TOT phenomena are driven by interference from competing non-target candidates within the language network.

The TOT phenomenon is a temporarily frustrating state in which individuals feel a strong familiarity with a word but struggle to produce it immediately ([16]). TOT states are often attributed to difficulties in retrieving phonological word-forms during speech production, typically linked to issues within the speaker’s lexical phonological network. However, conflicting views on the lexical phonological network’s role in TOT have led to two contrasting hypotheses: the blocking hypothesis and the transmission deficit hypothesis. The former predicts that a more enriched lexical phonological network, which means a larger amount of phonological neighbors for target words, leads to increased interference when retrieving the target words ([49]), whereas the latter predicts that a more enriched lexical phonological network would strengthen the activation of the target word, thereby reducing TOTs ([20], [19]; [45]). To address this controversy, the current study employed graph-theoretical metrics of the brain’s phonological processing network as proxies for individuals’ lexical phonological processing network, derived from DTI data in a large-scale cross-sectional cohort repository. Graph theoretical analysis of structural brain networks can provide insight into how the microstructural integrity of neural networks relates to cognitive processing ([10]; [30]).

At the global level, we found a positive correlation between global metrics (i.e., global efficiency and mean degree centrality) of the phonological processing network and TOT rates. Previous research has demonstrated that higher connectivity metrics within the phonological processing network are associated with a more enriched lexical network of phonological representations ([32]; [75]). Further studies have established a link between the white matter integrity of the language network and vocabulary size ([59]; [41]; [4]; [26]). Collectively, our findings suggest that a more enriched lexical network of phonological representations leads to the co-activation of a larger number of phonological neighbors, which interferes with the selection of target words and results in higher occurrences of TOTs. These findings support the blocking hypothesis and align with previous bilingual studies indicating that bilinguals exhibiting higher network efficiency in language networks typically experience more TOTs than monolingual individuals ([33]; [70]; [32]), likely due to additional phonological interference from other languages. Additionally, [85] ([85]) reported that bilinguals utilize similar phonological network structures when processing their native and second languages, indicating a high level of consistency in phonological processing mechanisms between bilinguals and monolinguals. Therefore, although the bilingual background of the participants in the present study is unknown, it is reasonable to suggest that our findings would not be significantly influenced by the participants’ language background.

At the nodal level, we found that the nodal efficiency of the bilateral pSTGs and the clustering coefficient of the left PMA positively predicted TOT rates. The bilateral pSTG is responsible for storing phonological representations ([66]), while the left PMA is involved in syllabification during speech production ([44]). Higher nodal metrics of these two regions indicate greater efficiency in facilitating the co-activation of a large number of phonological representations and/or syllabified segments. This can lead to increased interference with target words and result in more TOTs. In contrast, the clustering coefficient of the left pSMG and the degree centrality of the left dSTG negatively predicted tip-of-the-tongue occurrences. The left pSMG functions as a buffer for temporarily storing phonological representations, where competition among multiple phonological neighbors is resolved to select the target ([97]). The left dSTG plays a crucial role in error detection and self-monitoring during speech production ([13]; [39]). Higher network metrics in these two regions represent enhanced efficiency in their ability to resolve lexical competition and detect errors, leading to fewer TOTs.

The positive correlation between the topological properties of the phonological processing network and TOT rates can also be plausibly explained within the TDH framework through a lateral inhibition mechanism. According to [46] ([46]), this mechanism enables the most appropriate words to suppress less suitable candidates, which may lead to insufficient activation of the target word. Specifically, the more phonological neighbors present in the lexical network, the stronger the lateral inhibition effect, resulting in reduced activation of the target word and a higher occurrence of TOTs. Although the current evidence does not allow us to distinguish whether the insufficient activation of the target word is due to lateral inhibition or excessive activation of neighbors caused by interference leading to TOTs, we favor the blocking hypothesis. This is because lateral inhibition occurs only after the most activated word has been selected ([46]), whereas a TOT is a state in which no word can be selected.

Since age was a significant predictor in all regression models, we split the full sample into three age groups to examine detailed age differences. The results showed that some effects (e.g., global metrics and nodal metrics of the left PMA and pSMG) observed in the full sample analysis were present only in the elderly adult group while some other effects (e.g., the nodal metrics of the left dSTG and pSTG) were restricted in the young adult or middle-aged groups. However, despite these age-related effects, the predictors emerged as significant in the regression models, and the directions of their effects were highly consistent between the full sample analysis and the age-based group analysis. Thus, the results of the age-based group analysis did not change our conclusion.

This study has several limitations. First, while DTI offers valuable insights into the brain’s structural connectivity, it is inherently limited in its capacity to capture the dynamic complexity of brain function ([24]). In contrast, functional connectivity networks can reveal time-varying dynamics of the brain during specific tasks or states. Incorporating task-based fMRI along with advanced network metrics, such as dynamic functional connectivity ([42]) and dynamic causal modeling ([77]), can further characterize the time-varying network properties and interactions within the phonological processing network ([40]; [64]). Additionally, combining DTI-derived structural connectivity with task-based fMRI has proven effective in developing computational models of language production ([72]; [88]). For instance, [88] ([88]) implemented the Lichtheim 2 neurocomputational model—a model of speech production and comprehension—by constraining its dorsal and ventral pathways with DTI and task-evoked fMRI data, successfully simulating normal and aphasic language processes. Future research should aim to integrate both DTI and fMRI modalities to enhance these neurocomputational models of speech production. Second, although the ROIs used in the present study were selected based on meta-analyses of dozens of studies examining the neural underpinnings of phonological processing in language production ([39]; [43]; [44]; [51]; [66]), there remains ongoing debate within the literature regarding the precise delineation of the phonological processing network. Future studies should consider incorporating task-based fMRI, aimed at validating ROIs for phonological processing, within a large-scale network analysis. Third, the current study employed the classic public figure naming task to elicit TOT states. Proper names are commonly used for this purpose, as they are particularly susceptible to word retrieval failures due to having far fewer semantically related nodes ([20]). This unique property of proper names may result in their processing being different from that of other words. Future studies should employ alternative tasks and/or other lexicons to induce TOTs to yield a more comprehensive understanding of the cognitive neural mechanisms underpinning phonological processing.

In summary, the current results show that better microstructural integrity of the phonological processing network was associated with higher TOT rates. Our findings suggest that individuals with enriched phonological representations are more likely to experience TOT occurrences. This may be due to increased interference from target-related phonological neighbors, thereby supporting the blocking hypothesis of TOTs.

## Figures and Tables

**Figure 1 behavsci-15-00977-f001:**
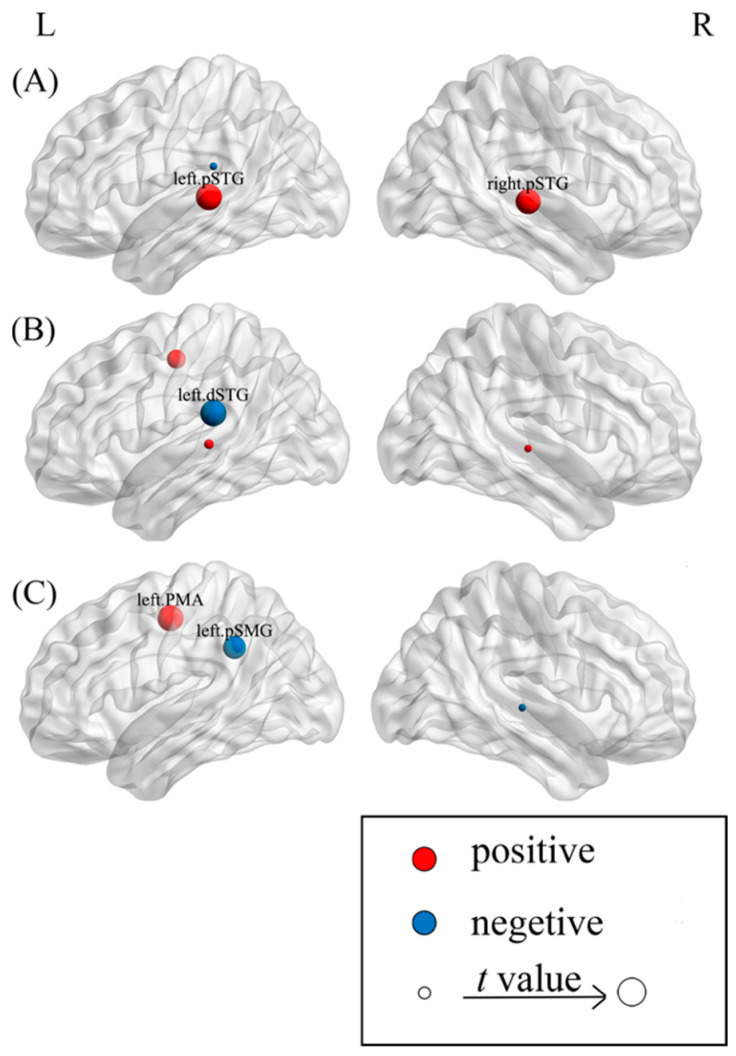
The relationship between nodal metrics and TOT rates. Note. (**A**–**C**) present the results of the stepwise regression models for nodal efficiency, degree centrality, and clustering coefficient in predicting TOT rates, respectively. Red nodes indicate positive correlations, while blue nodes represent negative correlations. The size of each node indicates t-values, with larger nodes signifying greater t-values. pSTG: posterior inferior frontal gyrus; dSTG: dorsal superior temporal gyrus; PMA: premotor area; pSMG: posterior supramarginal gyrus. L: left; R: right. Visualization was performed using the BrainNet Viewer ([95]).

**Table 1 behavsci-15-00977-t001:** HOA-112 coordinates of ROI.

Name	Coordinates		
Left premotor area (PMA)	−34.28	−11.71	49.18
Left posterior inferior frontal gyrus (pIFG)	−50.69	14.63	15.22
Left dorsal superior temporal gyrus (dSTG)	−48.4	−31.53	20.30
Left anterior insula (aINS)	−36.15	2.73	0.52
Left posterior superior temporal gyrus (pSTG)	−62.37	−29.14	3.86
Left posterior supramarginal gyrus (pSMG)	−54.86	−46.04	33.58
Right dorsal superior temporal (dSTG)	48.86	−27.69	21.65
Right posterior inferior frontal gyrus (pIFG)	61.35	−23.87	1.50

**Table 2 behavsci-15-00977-t002:** Results of regression models for phonological network in predicting TOT rates.

Model	Predictor	Standardized *β*	*t* Value	*p*
global efficiency	age	0.340	8.585	<0.001 ***
	gender	−0.026	−0.331	0.741
	global efficiency	0.093	2.320	0.021 *
local efficiency	age	0.340	8.542	<0.00 ***
	gender	−0.003	−0.033	0.974
	local efficiency	−0.021	−0.516	0.606
mean degree centrality	age	0.005	8.542	<0.001 ***
	gender	−0.024	−0.299	0.765
	mean degree centrality	0.077	1.925	0.055 ^†^
mean clustering coefficient	age	0.330	8.332	<0.001 ***
	gender	−0.003	−0.042	0.967
	mean clustering coefficient	0.018	0.448	0.654

Note: ^†^ < 0.07 * < 0.05 *** < 0.001.

**Table 3 behavsci-15-00977-t003:** Results of the stepwise regression analyses on each nodal metric and TOT rates.

Model	Predictor	Standardized *β*	*t* Value	*p*
nodal efficiency	age	0.332	8.319	<0.001 ***
	dSTG.l	−0.094	−1.771	0.077 ^†^
	pSTG.l	0.142	2.708	0.007 **
	pSTG.r	0.105	2.655	0.008 **
degree centrality	age	0.333	8.318	<0.001 ***
	dSTG.l	−0.114	−2.191	0.029 *
	PMA.l	0.107	1.928	0.054 ^†^
	pSTG.l	0.085	1.587	0.113
	pSTG.r	0.070	1.511	0.131
clustering coefficient	age	0.337	8.512	<0.001 ***
	PMA.l	0.171	3.091	0.002 **
	pSMG.l	−0.113	−2.814	0.005 **
	dSTG.r	−0.080	−1.429	0.154

Note: ^†^ < 0.07 * < 0.05 ** < 0.01 *** < 0.001.

**Table 4 behavsci-15-00977-t004:** Results of regression models for phonological network in predicting TOT ratio (different age groups).

Model	Age Group	Predictor	Standardized *β*	*t* Value	*p*
global efficiency	18–34	global efficiency	0.000	−0.004	0.997
	35–65	global efficiency	0.020	0.350	0.727
	66–87	global efficiency	0.179	2.756	0.006 **
local efficiency	18–34	local efficiency	−0.048	−0.715	0.477
	35–65	local efficiency	−0.052	−0.790	0.430
	66–87	local efficiency	0.174	2.126	0.035 *
mean degree centrality (DC)	18–34	mean DC	−0.038	−0.350	0.727
	35–65	mean DC	0.016	0.284	0.776
	66–87	mean DC	0.170	2.555	0.011 *
mean clustering coefficient (CC)	18–34	mean CC	−0.117	−0.350	0.727
	35–65	mean CC	0.050	0.284	0.776
	66–87	mean CC	0.521	2.555	0.011 *

Note: * < 0.05. ** < 0.01.

**Table 5 behavsci-15-00977-t005:** Results of the stepwise regression analyses on each nodal metric.

Model	Age Group	Predictor	Standardized *β*	*t* Value	*p*
nodal efficiency	18–35	dSTG.l	−0.292	−2.170	0.033 *
		pSTG.l	0.420	2.779	0.007 **
		gender	−0.327	−1.829	0.071
	36–65	pIFG.l	−0.088	−1.499	0.135
		pSTG.l	0.106	1.674	0.095
	66–87	PMA.l	0.195	2.860	0.005 **
degree centrality	18–35	dSTG.l	−0.226	−1.855	0.067 ^†^
		pSTG.l	0.265	2.077	0.041 *
		gender	−0.251	−1.421	0.159
	36–65	PMA.l	0.090	1.631	0.104
	66–87	PMA.l	0.187	2.796	0.006 **
clustering coefficient	18–35	pSTG.l	0.159	1.448	0.151
	36–65	dSTG.l	−0.132	−1.976	0.049 *
		PMA.l	0.208	2.523	0.012 *
		dSTG.r	−0.128	−1.594	0.112
	66–87	pIFG.l	0.163	1.781	0.076
		PMA.l	0.154	2.053	0.041 *
		pSMG.l	−0.222	−2.721	0.007 **

Note: ^†^ < 0.07 * < 0.05 ** < 0.01.

## Data Availability

Data used in the preparation of this study were obtained from the open-source Cambridge Center for Ageing and Neuroscience (Cam-CAN) data repository (https://cam-can.mrc-cbu.cam.ac.uk/dataset/) and we accessed to use the data on 30 August 2023.

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
