# Peer review of "The Relationship Between the Phonological Processing Network and the Tip-of-the-Tongue Phenomenon: Evidence from Large-Scale DTI Data"

_behavsci, 2025, doi:10.3390/bs15070977_

Round 1
Reviewer 1 Report
Comments and Suggestions for Authors
Thank you for submitting the manuscript. The authors investigate the tip-of-the-tongue (TOT) phenomenon by examining the relationship between phonological network integrity and TOT occurrences using diffusion tensor imaging (DTI) data from 576 participants. The study reveals that individuals with better microstructural integrity of the phonological processing networks experience more TOTs, supporting the blocking hypothesis for TOTs. A few considerations would strengthen the manuscript.
- Please explain the significance of white matter tracts in estimating functional connections through structural ones. Follow this with a detailed description of how a graph theoretical approach for the analysis of DTI can yield meaningful results pertaining to this study.
- Please discuss in greater detail advanced network metrics beyond the current graph theory approach to better characterize the time-varying network properties and interactions within the phonological processing network.
- Also discuss the possibility of computational modeling to simulate neural dynamics based on DTI-derived structural connectivity.
- Please expand the participant pool to include individuals with known cognitive impairments to explore ROI-specific perturbations.
- Please incorporate a wider range of word retrieval tasks beyond naming public figures. Additionally, incorporation of multilingual individuals would and choosing appropriate lexicon for the study would be also interesting.
- Please consider splitting the data into age groups and explore age-group related differences in the phonological processing network, beyond the simple treatment in this study.
Reviewer 2 Report
Comments and Suggestions for Authors
Dear authors,
I have read through the paper carefully and the results might be encouraging overall and of scientific interest. This paper gives an interesting view about the neural correlates underpinning the relationship between the phonological processing network and the Tip-of-Tongue (ToT) experience. Furthermore, the research offers a novel and interesting application of the graph analysis in order to investigate the extent to which the integrity of brain areas and their connections, which are involved in the processing network, affect the occurrence of the ToT phenomenon in healthy participants. However, I have both general and specific concerns, especially with the theoretical framing that I think they would need to address. I suggest major and minor revisions discussed below, which if addressed, could significantly improve the manuscript’s overall relevance and accessibility to a broader scientific audience.
Major revisions
In their paper, the authors compare two different theories underlying ToT experience, namely the blocking theory and the transmission deficit hypothesis. The authors posit that two different predictions could be made about the relationship between the phonological processing network and the occurrence of ToT. The predictions are tested through the application of the graph analysis applied to DTI data. However, I have some concerns in that regard. Firstly, the authors inferred that high integrity of the phonological processing network is associated with a higher number of phonological neighbours and vice versa. Accordingly, the positive correlation between the integrity of the phonological processing network and the ToT could offer evidence supporting the blocking theory. Reversely, the negative correlation could offer evidence supporting the transmission deficit theory. To my knowledge, there is no evidence that integrity of the phonological network may be conceived as a measure of the number of the phonological neighbours and the way they interact each other’s.
Secondly, the authors highlight that the transmission deficit hypothesis posit that retrieval failure occurs due to insufficient phonological activation of target words caused by weakened connections between the semantic nodes and the phonological nodes or connections within the phonological nodes. However, it would be relevant to flash out the ways the insufficient phonological activation may be due to different sources for example lateral inhibition mechanisms (see Warrington, 1975), which are not related to a small number of phonological neighbours or a reduced connection between them.
Finally, the comparison of the two models, and the support of one at expense of the other, need more evidence (e.g. other behavioural data) which are not provided in the present paper.
In conclusion, as far as I am concerned, the paper is of scientific relevance, but the overall framework should be revised. The entire manuscript should focus on the relationship between the phonological processing network and the ToT. The results may be discussed considering the two models in the relative section of the manuscript (i.e. Discussion).
Minor revision
- 35: the authors state: “frequently in elderly adults and individuals with conditions such as dyslexia or Parkinson’s disease”. People with aphasia should be reported along.
- 73 to 76: this paragraph should be more detailed.
In the results section, a brief discussion of the results should be reported. It would be beneficial to include a brief preliminary interpretation of the results. For example, the negative correlation observed between the left dorsal Superior Temporal Gyrus (dSTG) and the frequency of Tip-of-the-Tongue (TOT) states warrants further explanation. Providing an initial interpretation, such as the possibility that increased activation in the left dSTG may reflect more efficient phonological retrieval processes, would help orient the reader and underscore the significance of the finding.
Reviewer 3 Report
Comments and Suggestions for Authors
This manuscript investigates the relationship between phonological network structure and tip-of-the-tongue (TOT) states, using graph-theoretical metrics derived from diffusion tensor imaging (DTI) in a large sample (n = 576) from the Cam-CAN dataset. The study is framed as a test of two competing accounts: the blocking hypothesis, which attributes TOTs to interference from phonological neighbors, and the transmission deficit hypothesis, which views them as a failure of phonological activation. The finding that greater global efficiency and degree centrality in the phonological network predict higher TOT rates is interpreted as supporting the blocking account.
The aim is clear, and the analysis is generally well executed. The ROI selection is well justified, with references to prior meta-analyses and use of the Harvard-Oxford atlas to define regions such as the STG, pIFG, aINS, and pSMG. However, Table 1 contains a possible labeling inconsistency: the last row lists “Right posterior inferior frontal gyrus (pSTG),” which seems to conflate temporal and frontal anatomy. This should be clarified or corrected. Additionally, the formatting of the table is awkward. Minor edits would improve readability and presentation.
Several other areas would also benefit from revision prior to being accepted for publication. The core graph-theoretical measures—global efficiency, nodal efficiency, clustering coefficient, and degree centrality, are defined only through equations, with minimal conceptual explanation. Readers outside network neuroscience would benefit from brief, plain-language descriptions in the main text. Moreover, many equations extend beyond the page margins and need reformatting. The TOT task is described briefly, but it would be helpful to clarify how the TOT rate was computed and how responses were categorized. The discussion is long and at times repetitive—particularly in describing the functions of the pSTG and PMA—and would benefit from clearer structure organized around the main findings (e.g., global vs. nodal metrics).
While the results are consistent with the blocking hypothesis, the authors do not consider the possibility that both interference and transmission failure could contribute to TOTs under different circumstances. Bilingualism is mentioned in the introduction and discussion as a factor associated with increased TOTs and greater network efficiency, but the language background of participants is not reported. Given prior work linking bilingualism to both TOT frequency and white matter structure, this omission should be addressed. The authors also infer cognitive dynamics (e.g., activation, competition) based solely on structural connectivity measures. While this limitation is briefly noted, it deserves more emphasis.
Finally, the tone could be more precise in places. Phrases like “an annoying state” are too informal for a research context and should be replaced with more neutral, descriptive language such as “temporarily frustrating” or “retrieval-disruptive.” Overall, the manuscript raises a valuable question and is supported by solid data, but revisions are needed to improve clarity, tone, and coherence before it is ready for publication.
Round 2
Reviewer 2 Report
Comments and Suggestions for Authors
Dear authors, I am pleased to read the amended version of your manuscript. In my opinion, all the main concerns have been addressed. Furthermore, both comments and revisions provided have improved the manuscript’s overall relevance. In conclusion, I have no further concerns about the manuscript.